# Potential for Exposure to Particles and Gases throughout Vat Photopolymerization Additive Manufacturing Processes

**Lauren N. Bowers, Aleksandr B. Stefaniak \***, **Alycia K. Knepp, Ryan F. LeBouf, Stephen B. Martin, Jr.**, **Anand C. Ranpara, Dru A. Burns and M. Abbas Virji**

National Institute for Occupational Safety and Health, Morgantown, WV 26505, USA
\* Correspondence: astefaniak@cdc.gov; Tel.: +1-304-285-6302

**Abstract:** Vat photopolymerization (VP), a type of additive manufacturing process that cures resin to build objects, can emit potentially hazardous particles and gases. We evaluated two VP technologies, stereolithography (SLA) and digital light processing (DLP), in three separate environmental chambers to understand task-based impacts on indoor air quality. Airborne particles, total volatile organic compounds (TVOCs), and/or specific volatile organic compounds (VOCs) were monitored during each task to evaluate their exposure potential. Regardless of duration, all tasks released particles and organic gases, though concentrations varied between SLA and DLP processes and among tasks. Maximum particle concentrations reached 1200 $\#/cm^3$ and some aerosols contained potentially hazardous elements such as barium, chromium, and manganese. TVOC concentrations were highest for the isopropyl alcohol (IPA) rinsing, soaking, and drying post-processing tasks (up to 36.8 $mg/m^3$), lowest for the resin pouring pre-printing, printing, and resin recovery post-printing tasks (up to 0.1 $mg/m^3$), and intermediate for the curing post-processing task (up to 3 $mg/m^3$). Individual VOCs included, among others, the potential occupational carcinogen acetaldehyde and the immune sensitizer 2-hydroxypropyl methacrylate (pouring, printing, recovery, and curing tasks). Careful consideration of all tasks is important for the development of strategies to minimize indoor air pollution and exposure potential from VP processes.

**Keywords:** ultrafine particles; volatile organic compounds; tasks; 3-dimensional printing

## 1. Introduction

Additive manufacturing (AM) is the process of joining materials to make objects, usually one layer at a time, from 3-dimensional (3-D) model data [1]. As described in recent reviews, there are many studies on emissions, indoor air pollution, and exposure potential during AM printing in homes, schools, libraries, offices, and other spaces [2–6]; however, less attention has been given to understanding potential for indoor air pollution and exposure throughout an entire AM process, i.e., during pre-printing, printing, post-printing, and post-processing tasks [7,8]. Generally, most AM processes require specific tasks to be completed to ensure the quality of the final printed object. Although some tasks require very little operator interaction, others require direct handling by the worker, which can have exposure potential. These tasks include but are not limited to pre-printing (e.g., designing the 3-D model, preparing the print chamber, loading feedstock into the printer), printing (e.g., building the 3-D model layer-by-layer), post-printing (e.g., removing the as-built part, unloading and recycling excess feedstock from the printer), and post-processing (e.g., finishing as-built parts using a solvent, curing, sanding, or machining). Additionally, the magnitude of exposure potential does not necessarily increase with the length of time spent on various AM tasks [9]. For example, it was reported that operators spent comparatively little time near 3-D printers during the long-duration print task and their main exposure came when fully present during shorter duration post-processing tasks. Hence, understanding impacts on indoor air pollution and exposure potential throughout

an entire AM process is important for identifying and apportioning exposures to specific tasks so that targeted controls can be implemented, if necessary.

Vat photopolymerization (VP) is one type of AM process, in which a liquid photopolymer resin is selectively cured by light-activated polymerization to build an object [1]. Types of VP technology include, but are not limited to, stereolithography (SLA) and digital light processing (DLP). While both technologies use a light source to cure resin, the underlying principles of operation, and therefore, the duration of the printing task differ between them. SLA printers scan a laser beam across the print area to selectively cure the resin in a vat. The accuracy of the printed part is related to the diameter of the laser beam [10]. This process cures the resin as a series of points and rounded lines to build objects layer-by-layer. DLP printers use a high-resolution projector to flash black and white image slices of each object layer across the entire vat surface at once. The higher the projector resolution is, the more accurate the printed object will be [10]. The projector is a digital screen that forms white areas of the projected image made of square pixels that are then cured using multi-wavelength light from a lamp to build the object. Although SLA printers fabricate objects with higher accuracy than DLP, this increased resolution comes at a cost of longer production time [10].

The main components of photopolymer resins are binders (50–80%), monomers (10–40%), and photoinitiators (<10%) [11]. The release of some of these constituents into indoor air may be of health concern. For example, some resin binders include organic molecules such as acrylates and epoxies [11], which are known immune sensitizers [12–14]. Studies by Väisänen et al. indicated that specific acrylates released into the air during VP printing included 2-hydroxypropyl methacrylate, ethyl methacrylate, and methyl methacrylate [15,16]. A number of potentially toxic or reactive metals are used as photoinitiators, including antimony oxide, copper, zinc, and iron complexes, and titanium, aluminum, and barium compounds [11,17–20]. Despite the presence of known immune sensitizers and metals in feedstock resins, there is little understanding of release and potential exposure to these ingredients during specific VP process tasks. Hence, the purpose of this study was to evaluate airborne particle and organic chemical concentrations during pre-printing, printing, post-printing, and post-processing tasks to understand indoor air pollution and exposure potential throughout VP processes.

## 2. Materials and Methods

Five replicates (n = 5) were conducted for every task: pre-printing (resin pouring), printing, post-printing (resin recovery), and post-processing (isopropyl alcohol [IPA] rinsing, soaking and drying, and curing and sanding). One type of grey photopolymer resin (FLGPGR02 Formlabs Inc., Somerville, MA, USA) was used to perform the tasks utilizing one SLA printer (Form 1+, Formlabs Inc.) and one DLP printer (M-One, Makex Co., Ltd., Ningbo, Zhejiang, China).

### 2.1. Bulk Resin Characterization

Prior to performing simulated task assessments, the bulk feedstock resin was analyzed for elements and organic compounds to identify potential chemicals for air monitoring. Elemental content was determined in accordance with a modified National Institute for Occupational Safety and Health (NIOSH) Method 7303 by heated acid digestion and subsequent analysis by inductively coupled plasma mass-optical emission spectroscopy (ICP-OES) [21]. Briefly, 0.1 g of resin was placed into a pre-cleaned 50 mL polypropylene Questron® (Questron Technologies Corp., Mississauga, ON, Canada) digestion tube. The resin was digested by adding 2.5 mL of concentrated trace metal grade hydrochloric acid and heating the mixture to 95 °C for 15 min in a Questron® QBlock digestion block. The samples were removed and allowed to cool before adding 2.5 mL of concentrated trace metal grade nitric acid. Samples were heated to 95 °C again for 15 min, cooled, and diluted to a final volume of 50 mL. All digestions were conducted in triplicate. Elemental analysis was conducted on an Agilent 7900 ICP-OES (Agilent Technologies, Santa Clara, CA, USA).

Analysis of samples was conducted in no gas and collision mode using helium (5.0 mL/min). The composition of organic compounds in the resin was determined in accordance with EPA Method 8270 [22]. Briefly, 1 g of resin was dissolved in 16 mL of methanol in an amber vial, placed in an ultrasonic bath with ice for 20 min, and allowed to settle for one hour. All steps were performed in the dark to prevent photooxidation of resin constituents. Samples were analyzed using gas chromatography-mass spectrometry (GC-MS). Individual sample constituents were identified by comparing the mass spectral patterns and retention times of peaks in the samples to the mass spectral patterns and retention times of a standard curve (acetone, isopropyl alcohol, hexane, benzene, toluene, ethylbenzene, xylenes, and methyl acrylate, ethyl acrylate, ethyl methacrylate, 2-hydroxyethyl acrylate, 2-hydroxyethyl methacrylate, allyl methacrylate, glycidyl methacrylate, and hydroxypropyl methacrylate). Compounds in the standard curve were selected based on ingredient lists from Safety Data Sheets (SDS), prior characterization of vat resins (author's unpublished data), and available literature [5,12–16,23–26]. Additionally, tentatively identified compounds were qualitatively identified from the remaining chromatographic peaks based on the mass spectral matches with mass spectral library software (Wiley Registry 8th Edition/NIST 2008 Mass Spectral Library, Wiley-Blackwell, New York, NY, USA) and a manual review by the analyst.

### 2.2. Test Chambers

Figure 1 depicts the experimental setups (sample types and collection locations) used to evaluate potential exposures from the SLA and DLP processes. The resin pouring, printing, resin recovery, IPA rinsing and soaking, and air-drying tasks were performed in a temperature ($21 \pm 1$ °C) and humidity ($50 \pm 5\%$) controlled 12.85 m$^3$ stainless steel chamber [23,27]. Air entering the chamber was passed through carbon and high efficiency particulate air (HEPA) filters to remove organic gases and particles, respectively. The ultraviolet curing task (described below) was performed in a 0.5 m$^3$ stainless steel chamber with carbon- and HEPA-filtered makeup air. The sanding task was performed inside a 0.065 m$^3$ spherical polycarbonate glove box enclosure (Techni-Dome® 360 Glove Chamber, Bel-Art Products, Inc., Wayne, NJ, USA) with carbon- and HEPA-filtered makeup air. Using sulfur hexafluoride, the air exchange rates for the 12.85 m$^3$ stainless steel chamber and the 0.5 m$^3$ stainless steel chambers were 1 per hour and 1.4 per hour, respectively. Based on the sampling rate of all instruments, the air exchange rate for the 0.065 m$^3$ glove box was calculated to be 13.7 per hour.

### 2.3. Air Monitoring

Separate air monitoring was performed before each task run to establish background levels of particles and gases in chambers and during each task run (start of task through completion plus time to follow the decay of contaminant levels) (see Table 1). Real-time instruments and time-integrated sampling approaches were used to measure particle- and gas-phase contaminant levels; the combinations of approaches were tailored to each specific task and are given in the Section 3.

Total particle number concentration from 0.02 to 1 μm (P-Trak condensation nuclei counter [CNC], TSI Inc., Shoreview, MN, USA) and number concentration and size distribution of particles from 5.6 to 560 nm (fast mobility particle sizer [FMPS], TSI Inc.) or from 0.5 to 20 μm (aerodynamic particle sizer [APS], TSI Inc.) were monitored in real-time. Airborne particles were collected onto 37 mm diameter, 3.0 μm pore size track-etched polycarbonate (TEPC) filters by drawing chamber air through filters at 5 L/min using a pre-calibrated sampling pump (AirChek XR 5000, SKC Inc., Eighty Four, PA, USA). TEPC filters were analyzed in-house using a field emission scanning electron microscope (FE-SEM, S-4800, Hitachi, Tokyo, Japan) to determine particle morphology, and energy dispersive x-ray analysis was used to identify elemental constituents. Additionally, particles were collected using mixed cellulose ester (MCE) filters (0.8 μm, 37 mm in 3-piece clear plastic cassettes, SKC Inc.) at 5 L/min using a pre-calibrated sampling pump (AirChek XR5000,

SKC Inc.) and analyzed for elemental content by ICP-OES in accordance with NIOSH Method 7303 [21]. Air sampling results were compared to the elemental content of the bulk resin (see Supplemental Table S1) and literature information on photoinitiators.

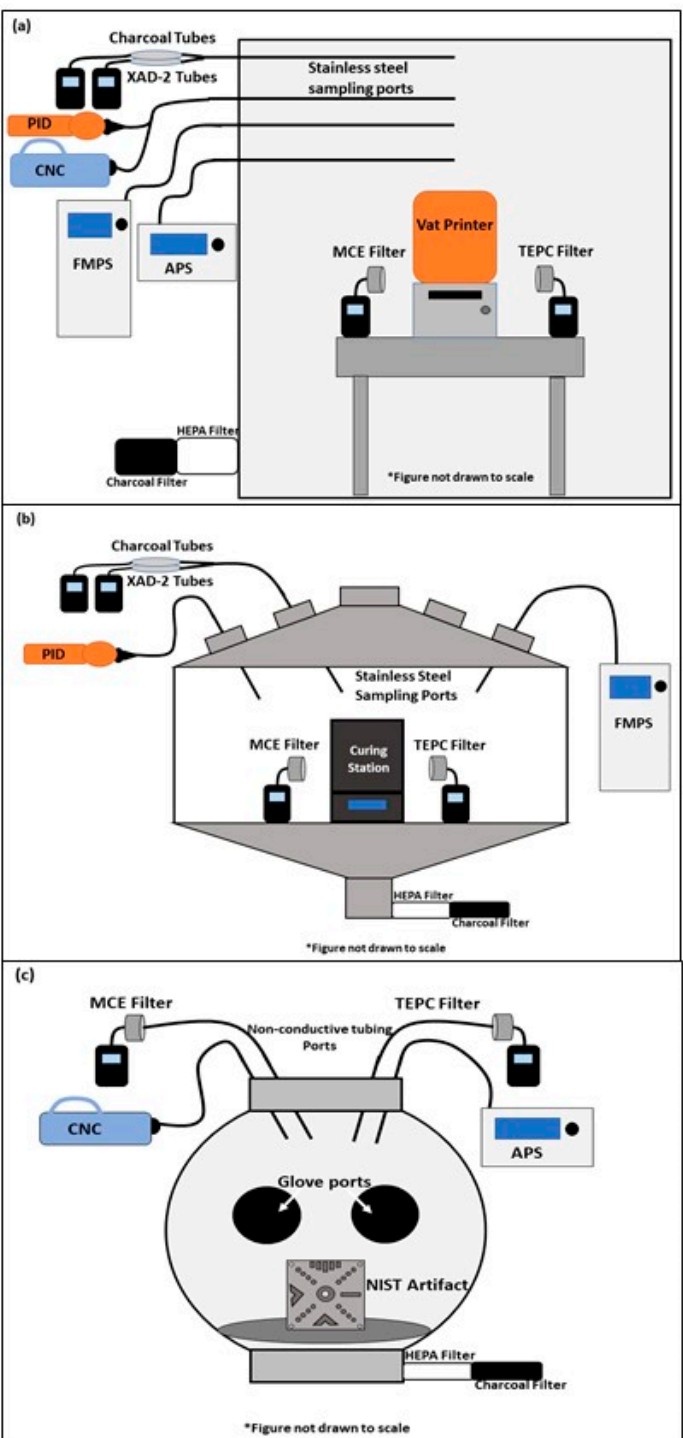

**Figure 1.** (a) 12.85 m³ chamber utilized for the pouring, printing, recovery, and isopropyl alcohol rinsing, soaking, and drying tasks; (b) 0.5 m³ chamber used for the curing task; (c) 0.065 m³ glove box used for the sanding task. Sampling ports leading into each chamber connected to specific real-time instruments and sampling media tailored for each task. APS = aerodynamic particle sizer; CNC = condensation nuclei counter; FMPS = fast mobility particle sizer; HEPA = high efficiency particulate air; PID = photoionization detector; MCE = mixed cellulose ester; TEPC = track-etched polycarbonate.

**Table 1.** Durations of tasks and decay periods performed during two types of vat polymerization additive manufacturing processes [1].

| Task | SLA (minutes) | DLP (minutes) | Decay (minutes) |
|---|---|---|---|
| Pouring (resin) | 0.5–0.7 | 0.5–0.8 | 60 |
| Printing (resin) | 145 | 163 | 60 |
| Recovery (resin) | 1–2 | 1–2 | 60 |
| Rinsing (IPA) | 2 | 1 | n/a |
| Soaking (IPA) | 20 | 20 | n/a |
| Drying (IPA) | 90 | 90 | 90 |
| Curing (resin and IPA) | 44 | 44 | 180 |
| Sanding | 3 | 3 | 60 |

[1] DLP = digital light processing, IPA = isopropyl alcohol, SLA = stereolithography, n/a = not applicable as the task previously performed immediately initiated the next task.

Total volatile organic compound (TVOC) concentrations were monitored using a real-time photoionization detector (PID) (Ion Science Inc., Stafford, TX, USA) calibrated with isobutylene. Samples for specific VOCs (C1 to C10) were collected during task simulations using whole-air 450 mL Silonite®-coated canisters (Entech Instruments, Inc., Simi Valley, CA, USA) followed by off-line analysis in-house by GC-MS [28]. Different flow controllers were used depending on the duration of the task for adequate sampling. For the background collection, a flow rate of 10 mL/min was used; for pouring and recovering a flow rate of 0.83 mL/min was used; for printing and curing a flow rate of 0.42 mL/min was used. Additional air samples were collected for select semi-volatile organic compounds based on the results of the bulk resin analysis and the ingredients listed on the resin product SDS. Based on the results of the bulk resin analysis, 2-hydroxyethyl acrylate, 2-hydroxyethyl methacrylate, and 2-hydroxypropyl methacrylate were sampled using charcoal sorbent tubes (6 × 70-mm size, 2 sections, 50/100 mg sorbent, 20/40 mesh, SKC Inc.) and pre-calibrated sampling pumps (Pocket Pump 210-1002MTX, SKC Inc.) operated at 0.20 L/min and analyzed by GC-MS based on Occupational Safety and Health Administration Method PV2252 [29]. The resin SDS listed methacrylated oligomer and monomer as ingredients, though only methacrylate monomer was identified by the bulk resin analysis. To evaluate if the resin released methacrylate monomer in the form of methyl or ethyl methacrylate, air samples were collected on sorbent tubes containing XAD-2 resin (8 × 110-mm size, 2 sections, 200/400 mg sorbent, 20/60 mesh, SKC Inc.) using pre-calibrated sampling pumps (Pocket Pump 210-1002MTX, SKC Inc.) operated at 0.05 L/min and analyzed by GC-flame ionization detection in accordance with NIOSH Method 2537 [30].

*2.4. Task Descriptions*

Eight sequential tasks in the VP process were monitored five times for each type of printer. The first 6 tasks were performed with the printers inside the 12.85 $m^3$ chamber. Task 7 was performed in the 0.5 $m^3$ chamber and task 8 was performed in the 0.065 $m^3$ glove box.

1. Pouring: Before printing, the resin bottle was mixed thoroughly by shaking vigorously. The bottle was uncapped, the printer cover was opened, and the resin was poured into the vat until the level reached the manufacturer-specified maximum fill line. Immediately after pouring, the printer cover was closed, and the bottle was recapped and removed from the chamber.

2. Printing: An artifact from the National Institute of Standards and Technology (NIST) was printed at a 50% scale of the full size [31]. The layer height of the NIST file for the SLA printer was 0.05 mm (PreForm v2.19.3, Formlabs Inc.) and 0.015 mm for the DLP printer (XMaker v2.5.4, Makex Co., Ltd.), which were the software recommended settings for each printer. Supports for the part were automatically generated in each software. The specific software meant for this DLP printer required manual input of four specific exposure times (raft [120 s], base [40 s], model [20 s], and darktime [2 s]). The raft is the layer that adheres the part to the build plate; the base is the layers of support between the raft and the model; the model is the remaining

layers of the print job. Darktime refers to the build plate being raised and lowered after the completion of each layer. These exposure times were chosen based on the manufacturer's recommended settings as well as trial and error. The lid on both printers must be closed in order for the printer to operate

3. Recovery: After printing, any unused resin was recovered from the vat from each machine and saved for reuse. To perform this post-printing task, the printer cover was opened, and the vat containing unused resin was removed from the printer. The resin was slowly poured into a 250 mL polypropylene bottle that was wrapped in foil to prevent exposure to light, and using a putty knife provided by the manufacturer, the vat was gently scraped to remove as much resin as possible. The printer cover was closed and remained in the chamber, and vat and the resealed bottle that contained recovered resin were immediately removed from the chamber and stored in the dark. The printed part remained attached to the build plate, and a paper towel replaced the vat to prevent any interior damage from resin dripping.

4. Rinsing: Built objects were subjected to sequential post-processing tasks. The first step was an IPA rinse. According to the printer manufacturers, for the SLA machine, the rinse task uses two containers whereas, for the DLP machine, the rinse task uses one container. For consistency, one container was used for both the SLA and DLP machines. The printer cover was opened to remove the build plate so the printed part could be removed. The printer cover was then closed (without the build plate), and the printed object was placed in a container with 500 mL of IPA (99.9%, HPLC Grade, Thermo Fisher Scientific, Waltham, MA, USA), the lid sealed, and the container was shaken for 1 to 2 min.

5. Soaking: After rinsing, objects sat in their sealed containers without agitation to soak in IPA.

6. Drying: The object was removed from the IPA container and placed on paper towels with the lights turned off in the chamber to air dry overnight.

7. Curing: To finish the hardening (curing) of any under- or un-cured resins on the objects, they were treated with ultraviolet (UV) radiation and heat [32]. A commercially available UV curing chamber (Form Cure, Formlabs Inc.) was used to cure all printed objects. Build supports on the printed objects were removed with flush cutters before placing the objects inside the curing chamber. Each printed object was subjected to UV light at 405 nm in a 60 °C air atmosphere.

8. Sanding: Printed objects were sanded using a sequence of three different grit sizes to give the finished product a cleaner and more professional look. Coarse grit (220) sandpaper was used to sand the surface where the supports were connected. Medium grit (660) and fine grit (1000) sandpaper were used to finish the part until it was smooth to the touch. Sanding took one minute for each grit with 10-min intervals between sanding to allow for airborne contaminant concentrations in the chamber to decay.

*2.5. Statistical Analysis*

The average particle (#/cm$^3$) and vapor (µg/m$^3$) concentrations for each task were calculated for the duration of the performed task. These average values were then background-corrected by subtracting the one-minute average before the task was performed. If results from one (20% censored) or two (40% censored) of five runs of real-time monitor data (average or maximum values) or time-integrated sampling results were below zero after background-correction or below an analytical limit of detection (LOD), respectively, then the β-substitution method was used to impute these values [33]. Briefly, this method calculates an adjustment factor (β) from the uncensored data in a dataset for the given task and printer combination. The β factor was multiplied by the mean background for a task and printer combination or a LOD to calculate a substitution value. Inspection of normal-probability plots indicated that real-time summary values and the time-integrated sampling results were more normally distributed when log-transformed. Hence, natural

log-transformed uncensored and β-substituted values were used to calculate geometric means (GM) and geometric standard deviations accounting for repeated measures using JMP software (version 13.0.0, SAS Institute Inc., Cary, NC, USA). Natural log-transformed real-time and time integrated sampling results were modeled using linear regression with a random effect for the repeated measures, and specific pairwise differences between printer technologies and tasks (fixed effects) were compared using Student's *t*-test and Tukey's HSD test as appropriate in JMP. A significance level of α = 0.05 was used for all comparisons.

## 3. Results

Table 1 summarizes the duration of each task by VP process. The pouring, recovery, rinsing, and sanding tasks each required less than three minutes, the soak and cure tasks required approximately 20 to 45 min, and the dry and print tasks lasted approximately 90 to 160 min, respectively.

Table 2 summarizes the GM average real-time monitoring data by process and task. In general, average particle number (#/cm$^3$) concentrations by task were often similar or below background levels. Average TVOC (μg/m$^3$) concentrations were well above the background for the rinsing, soaking, drying, and curing tasks.

**Table 2.** Geometric mean (geometric standard deviation) average background-corrected airborne particle and total volatile organic compound (TVOC) concentrations and particle sizes by process and task (n = 5 replicates) [1].

| | | Particles | | | | | Gases |
|---|---|---|---|---|---|---|---|
| | | CNC | FMPS | | APS | | PID |
| Task | Process | Conc. (#/cm$^3$) * | Conc. (#/cm$^3$) * | Size, nm (GSD) ‡ | Conc. (#/cm$^3$) * | Size, nm (GSD) ‡ | TVOC (μg/m$^3$) * |
| Pouring (resin) | SLA | 44.0 (2.0) [A,†] | 14.1 (2.2) [A,†] | 51.4 (1.3) | 0.1 (2.9) [B] | 678.2 (1.3) | – |
| | DLP | – | – | 55.4 (1.3) | – | 696.6 (1.4) | 1.1 (1.8) [D] |
| Printing (resin) | SLA | – | – | 40.6 (1.5) | – | 658.2 (1.2) | 61.0 (1.7) [D] |
| | DLP | – | – | 33.3 (1.5) | – | 654.4 (1.3) | – |
| Recovery (resin) | SLA | – | – | 41.0 (1.4) | 0.1 (1.9) [B,‡] | 801.0 (1.6) | – |
| | DLP | 7.9 (1.6) [A] | – | 29.8 (1.5) | 0.2 (1.8) [B,‡] | 876.2 (1.6) | 0.4 (4.3) [C,D,†] |
| Rinsing (IPA) | SLA | – | 10.5 (3.5) [A,&] | 44.2 (1.5) | 0.1 (1.4) [B,†] | 779.8 (1.5) | 1042.5 (4.4) [C] |
| | DLP | 8.8 (2.7) [A] | 31.7 (3.1) [A,‡] | 29.6 (1.6) | 0.3 (1.1) [B,‡] | 831.8 (1.6) | 0.4 (2.6) [C,†] |
| Soaking (IPA) | SLA | 20.7 (1.7) [A] | – | 48.6 (1.4) | 0.1 (1.8) [B] | 769.4 (1.5) | 8080.7 (1.9) [A] |
| | DLP | 27.0 (1.9) [A] | – | 33.1 (1.5) | 0.5 (3.0) [B,‡] | 764.4 (1.5) | 15,176.8 (1.6) [A] |
| Drying (IPA) | SLA | – | – | 49.1 (1.5) | – | 695.2 (1.3) | 8322.3 (1.5) [A] |
| | DLP | – | – | 39.4 (1.6) | – | 726.2 (1.4) | 13,384.9 (1.3) [A] |
| Curing | SLA | n/a | – | 36.2 (1.3) | n/a | n/a | 1867.2 (1.0) [A,B] |
| | DLP | n/a | – | 44.3 (1.4) | n/a | n/a | 1654.2 (1.1) [A,B] |
| Sanding | SLA | – | n/a | n/a | 4.6 (1.9) [A,‡] | 1334.8 (1.9) [A] | n/a |
| | DLP | – | n/a | n/a | 0.3 (4.7) [A,‡] | 1024.5 (1.7) [B] | n/a |

[1] CNC = condensation nuclei counter, FMPS = fast mobility particle sizer, GSD = geometric standard deviation of diameter, APS = aerodynamic particle sizer, PID = photoionization detector, TVOC = total volatile organic compounds. * Adjusted regression models (differences among tasks are the same for both printers and there is no difference between printers for each task). Within each column, capital letters for GM values that are not connected by the same letter are significantly different (*p* < 0.05). For example using the PID data, the "Drying [A]" and "Soaking [A]" tasks are both significantly different than the "Rinsing [C]" task. ‡ Mixed regression models (significant interaction between printer and task). FMPS size data (all *p* < 0.05): DLP process—pouring > printing or recovery or rinsing or soaking or drying tasks. APS size data (all *p* < 0.05): SLA process—sanding > all other tasks; DLP process—sanding > pouring = printing = drying; between processes—SLA sanding > DLP sanding. – = three or more measurements during task were below background (GM not calculated). ‡ One of five replicate values imputed using β-substitution method (see Method section). † Two of five replicate values imputed using β-substitution method (see Method section). & One of four replicate values (data for fifth replicate loss because of instrument error) imputed using β-substitution method (see Method section) n/a = task not monitored with this instrument.

Table 3 gives the GM maximum particle number and TVOC concentrations. Maximum values exceeded background levels for all tasks, except the pouring task APS data for the DLP process.

**Table 3.** Geometric mean (geometric standard deviation) maximum background-corrected particle and total volatile organic compound (TVOC) concentrations by process and task (n = 5 replicates).

| | | Particles | | | Gases |
|---|---|---|---|---|---|
| | | CNC | FMPS | APS | PID |
| Task | Process | Conc. (#/cm$^3$) * | Conc. (#/cm$^3$) * | Conc. (#/cm$^3$) ‡ | TVOC (µg/m$^3$) ‡ |
| Pouring (resin) | SLA | 57.9 (2.3) [B,C] | 325.5 (2.9) [B] | 0.2 (2.2) | 3.6 (1.8) [ǂ] |
| | DLP | 39.3 (1.7) [B,C] | 197.9 (2.7) [B] | – | 5.5 (1.2) |
| Printing (resin) | SLA | 125.0 (1.4) [A] | 907.2 (1.4) [A] | 0.3 (1.1) | 140.7 (1.5) |
| | DLP | 270.5 (1.5) [A] | 1085.2 (1.7) [A] | 2.7 (4.2) | 11.2 (1.4) |
| Recovery (resin) | SLA | 35.2 (1.6) [B,C] | 456.8 (1.6) [A,B,!] | 0.1 (2.0) | 1.8 (2.0) |
| | DLP | 52.2 (1.4) [B,C] | 790.3 (1.7) [A,B] | 0.4 (1.7) [ǂ] | 4.3 (1.9) |
| Rinsing (IPA) | SLA | 28.2 (1.4) [C,ǂ] | 528.8 (1.9) [A,B,!] | 0.2 (1.5) [ǂ] | 9113.0 (2.9) |
| | DLP | 38.9 (1.6) [C] | 685.1 (3.7) [A,B] | 0.7 (1.7) [ǂ] | 6.0 (1.5) |
| Soaking (IPA) | SLA | 80.4 (1.2) [A,B] | 554.4 (1.5) [A,B,!] | 0.3 (1.8) | 22,890.2 (2.2) |
| | DLP | 97.9 (1.4) [A,B] | 623.5 (2.5) [A,B] | 1.1 (2.6) [ǂ] | 36,790.9 (2.0) |
| Drying (IPA) | SLA | 98.0 (1.5) [A] | 1206.0 (1.7) [A,!] | 0.3 (2.1) | 19,443.5 (1.4) |
| | DLP | 197.6 (2.2) [A] | 1010.8 (1.3) [A] | 3.0 (6.1) | 33,417.3 (1.6) |
| Curing | SLA | n/a | 730.8 (1.8) [A] | n/a | 3083.4 (1.0) |
| | DLP | n/a | 869.6 (1.4) [A] | n/a | 2990.0 (1.0) |
| Sanding | SLA | 57.3 (1.8) [B,C] | n/a | 8.3 (1.9) | n/a |
| | DLP | 51.1 (3.0) [B,C] | n/a | 0.6 (4.3) [ǂ] | n/a |

* Adjusted regression models (differences among tasks are the same for both printers and there is no difference between printers for each task). Within each column, capital letters for GM values that are not connected by the same letter are significantly different ($p < 0.05$). For example using CNC data, the "Drying [A]" task is significantly different than the "Recovery [B,C]" and "Sanding [B,C]" tasks. ‡ Mixed regression models (significant interaction between printer and task). APS concentration data: SLA process—sanding > all other tasks; DLP process—printing = drying > pouring; between processes—DLP printing > SLA printing, DLP drying > SLA drying, and SLA sanding > DLP sanding. PID (TVOC) concentration data: SLA process—soaking = drying > curing > printing > pouring = recovery and rinsing = curing > pouring = printing = recovery; DLP process—soaking = drying > curing > pouring = printing = recovery = rinsing; between processes—SLA rinsing > DLP rinsing and SLA printing > DLP printing. – = three or more measurements during task were below background (GM not calculated). ǂ One of five replicate values imputed using β-substitution method (see Method section). ! Four replicate measurements (data for fifth replicate lost because of instrument error). n/a = not applicable (task not monitored with this instrument).

As summarized in Table 4, airborne concentrations of 2-hydroxypropyl methacrylate and several VOCs varied between VP processes and among the pouring, printing, recovery, and curing tasks within a VP process. Additional specific VOC results are given in the Supplemental File.

Five elements—arsenic (As), tin (Sn), titanium (Ti), zinc (Zn), and zirconium (Zr)—were detected in both the bulk resin and in aerosol collected on filter samples during resin-associated tasks (see Supplemental Table S2). Additional elemental sampling results are given in the Supplemental File.

All results reported in the main text and statistical analyses are based on the ln-transformed data. Bar graphs based on the untransformed data are presented in the Supplemental File (see Figures S1–S3) to illustrate measures by task.

**Table 4.** Background-corrected airborne concentrations of organic gases for resin-associated tasks. Values are geometric mean and (geometric standard deviation) unless noted otherwise (n = 5 replicates) [1].

| Task | Process [2] | 2-Hydroxypropyl Methacrylate * ($\mu g/m^3$) | Acetaldehyde ($\mu g/m^3$) | Acetone ($\mu g/m^3$) | Ethanol ‡ ($\mu g/m^3$) | Methylene Chloride ($\mu g/m^3$) | Styrene ($\mu g/m^3$) |
|---|---|---|---|---|---|---|---|
| Pouring (resin) | SLA | 50.7 | 14.4–18.2 † | 2.1–26.0 ⫫ | 3.6–14.0 ⫫ | 23.5–76.0 ⫫ | <1.52 |
| | DLP | <43.4 | 18.6–20.6 | 21.1 (1.4) | 6.7–26.2 ⫫ | 6.2 (2.5) | 4.1–8.9 |
| Printing (resin) | SLA | 18.1–21.9 | 10.0–16.3 | 2.7–10.1 † | 0.02–4.5 † | 0.03–16.0 ⫫ | <1.52 |
| | DLP | 31.2 (1.5) A,B | 8.3 (1.4) | 13.6 (1.5) | 7.4 (3.2) | 3.4 (5.4) | 4.5 |
| Recovery (resin) | SLA | 43.8 (1.2) A,† | 19.7–35.4 | 22.8 (1.9) | 14.2 (3.8) ⫫ | 0.69–38.6 ⫫ | <1.52 |
| | DLP | 58.5 (2.8) A,† | 10.9 | 17.5 (1.5) | 9.7 (2.3) | 5.8 (1.9) | 17.6 |
| Curing | SLA | 11.3 | 0.3–15.7 ⫫ | 15.9–23.6 | 4.6 (2.0) ⫫ | 3.9 | <1.52 |
| | DLP | 25.9 (1.8) B | 4.5–9.0 | 2.8–11.1 † | 0.6 (9.4) † | <0.204 | 2.0–4.5 |

[1] < = all samples below limit of detection, single value = one sample result above background, range = two or three sample results above background. [2] SLA = stereolithography, DLP = digital light processing. * Adjusted regression models (differences among tasks are the same for both printers and there is no difference between printers for each task). Within each column, capital letters for GM values that are not connected by the same letter are significantly different ($p < 0.05$). For example, using 2-Hydroxypropyl methacrylate data, the "Recovery A" task is significantly different than the "Curing B" task. ‡ Mixed regression models (significant interaction between printer and task). SLA process—recovery > printing ($p < 0.05$); DLP process—recovery > curing ($p < 0.05$). ⫫ One of five replicate values imputed using β-substitution method (see Method section). † Two of five replicate values imputed using β-substitution method (see Method section).

### 3.1. Pouring

Among real-time measurements, maximum particle number concentration (APS data) and maximum TVOC concentration were significantly different between the SLA and DLP processes (see Table 3). 2-Hydroxypropyl methacrylate was observed in one air sample during pouring into the SLA vat (50.7 $\mu g/m^3$) but was below the LOD during pouring into the DLP vat (see Table 4). Several VOCs were quantified during pouring for both VP processes (see Table 4), including acetaldehyde, acetone, ethanol, and methylene chloride. Chromium (Cr), Sn, Ti, Zn, and Zr were quantified on at least one air sample from the SLA and DLP processes during this task (see Supplemental Table S2). Additionally, iron (Fe) and manganese (Mn) were quantified on at least one sample for the SLA process, and barium (Ba), cadmium (Cd), molybdenum (Mo), strontium (Sr), and vanadium (V) on at least one sample for the DLP process. Supplemental Figure S4 shows examples of the morphology of some airborne particles released during resin pouring. Though carbon was ubiquitous in all samples because the filter was carbon-based, corresponding EDX spectra of the particles identified carbon. EDX analysis of other particles for the SLA process identified Cr and magnesium (Mg), and for the DLP process some particles contained aluminum (Al), Cr, and Mg (data not shown).

### 3.2. Printing

Maximum particle (CNC and FMPS data) concentrations were not significantly different between SLA and DLP printing. Maximum particle concentrations measured using an APS were just slightly above the background, though the GM concentration for the DLP printer was significantly higher compared with the SLA printer; $p < 0.05$ (see Table 3). Maximum TVOC concentration was significantly higher for the SLA printer (GM = 140.7 $\mu g/m^3$) compared with the DLP printer (GM = 11.2 $\mu g/m^3$); $p < 0.05$ (see Table 3). 2-Hydroxypropyl methacrylate was quantified in the air during printing with both types of printers (SLA, range = 18.1–21.9 $\mu g/m^3$; DLP, GM = 31.2 $\mu g/m^3$). Acetaldehyde, acetone, ethanol, methylene chloride, and styrene were quantified during some, but not all, print runs for these processes (see Table 4). Concentrations of Cr, Fe, Mn, and Zn in airborne particles exceeded background during SLA printing, whereas only concentrations of Ba and Zn exceeded background during DLP printing (see Supplemental Table S2). Lead (Pb) was quantified in

aerosol released during both the SLA and DLP processes and Sr during DLP printing only (see Supplemental File). Examples of particle morphologies observed during printing with the SLA and DLP printers are shown in Supplemental Figure S5. Carbon was present in all particles, and Cr and sodium (Na) in particles from DLP printing. Inspection of other particles identified Al, Cr, Fe, Mg, and Ti during printing with the SLA machine and Al, Cr, Fe, and Mg during printing with the DLP machine (data not shown).

*3.3. Recovery*

None of the real-time particle or TVOC metrics were significantly different between the SLA and DLP processes (see Tables 2 and 3). GM levels of 2-hydroxypropyl methacrylate were 43.8 µg/m$^3$ and 58.5 µg/m$^3$, for the SLA and DLP processes, respectively (not significantly different). From Table 4, specific VOCs that were quantified during this task included acetaldehyde, acetone, ethanol, and methylene chloride. Levels of Cr, Ti, Zn, and Zr in aerosols exceeded the background during the recovery task for both the SLA and DLP processes. Other elements quantified in aerosol on at least one sample the SLA process included Fe, Mn, and Pb, and the DLP process included As, Cd, Mo, V, and tellurium. Some particles released during resin recovery had nanoscale agglomerate morphology and were composed of carbon (see Supplemental Figure S6); other particles released during the recovery task contained Al and Fe (SLA process) and Fe, Mg, and Ti (DLP process).

*3.4. Rinsing*

Among all real-time measurements, only maximum TVOC concentration (see Table 3) was significantly different between the rinse task for NIST artifacts made with the SLA process (GM = 9113.0 µg/m$^3$) compared with the DLP process (6.0 µg/m$^3$); $p < 0.05$.

*3.5. Soaking*

There were no statistical differences in average or maximum particle and TVOC concentrations for the soaking task between the SLA and DLP processes.

*3.6. Drying*

Among maximum number concentration measurements, only data from the APS was significantly higher for the dry task with NIST artifacts made using the DLP printer compared with artifacts made using the SLA printer ($p < 0.05$), albeit at a level just above the background. There were no statistical differences in average or maximum TVOC concentrations for the drying task between the SLA and DLP processes.

*3.7. Curing*

There was no statistical difference in maximum particle concentration (FMPS data) or average and maximum TVOC concentrations for the curing task with objects made using the SLA and DLP processes. 2-Hydroxypropoyl methacrylate was quantified on just one charcoal tube sample for the cure task with objects made using the SLA printer (11.3 µg/m$^3$) but on all five samples for the objects made on the DLP printer (GM = 25.9 µg/m$^3$). Acetaldehyde, acetone, and ethanol were quantified on most samples during the curing task for the SLA and DLP processes (see Table 4). Elemental analysis of filter samples collected during the curing task for objects made using the SLA printer quantified Cr, Fe, Mn, Sn, Ti, Zn, and Zr above background, whereas, for objects made using the DLP printer, aerosolized elements included Fe, Mn, Sn, Zn, and Zr (see Table S2). Some particles released during curing had branch-chain morphology and were composed of carbon (see Supplemental Figure S7). Evaluation of other particles released during this task identified Fe in aerosol from objects made using the SLA printer and Fe and nickel (Ni) from objects made using the DLP printer (data not shown).

### 3.8. Sanding

Maximum particle concentration (APS data) was just above background, but significantly higher when sanding objects made using the SLA printer (GM = 8.3 #/cm$^3$) compared with objects made using the DLP printer (GM = 0.6 #/cm$^3$); $p < 0.05$ (see Table 3). Additionally, particle size (APS data) was significantly larger during the sanding of objects made using the SLA printer (GM = 1.34 μm) compared with objects made using the DLP printer (GM = 1.03 μm); $p < 0.05$ (see Table 2). Objects were sanded using progressively finer sandpaper grit; however, there was no statistical difference in aerosol concentrations by grit (data not shown). Aerosol released during sanding contained Ba, Cr, Fe, Mn, Sn, Zn, and/or Zr (see Table S2). From Supplemental Figure S8, some aerosol particles generated during sanding were microscale particles composed of carbon. Additionally, Al was observed in some particles released during sanding of objects made using the SLA printer and Fe and Ni were present in some particles released during sanding of objects made using the DLP printer (data not shown).

### 3.9. Relationships among Tasks

For total particle number concentration from 0.02 to 1 μm (CNC data) there were no statistical differences in GM average particle concentrations between printers or among tasks (see Supplemental Table S2). In contrast, significant differences were observed among tasks for GM maximum number concentrations (see Table 3). Specifically, for both the SLA and DLP processes, the same orders of maximum number concentrations from the CNC data were observed (from highest to lowest): printing = drying > pouring = recovery = rinsing = sanding ($p < 0.05$) and soaking > rinsing ($p < 0.05$).

For the number of concentration and size distribution of particles from 5.6 to 560 nm (FMPS data), average concentration values (see Table 2) were generally similar to the background. Maximum concentrations (see Table 3) for the drying, printing, and curing tasks were significantly higher compared with the pouring task ($p < 0.05$). From Table 2, for the SLA process, there were no statistical differences in GM particle sizes among tasks. For the DLP process, GM particle size for the pouring task (55.4 nm) was significantly larger than for the printing (33.3 nm), recovery (29.8 nm), rinsing (29.6 nm), soaking (33.1 nm), or drying (39.4 nm) tasks; $p < 0.05$.

For the largest aerosol size fraction, from 0.5 to 20 μm (APS data), average number concentrations were just above the background (see Table 2), though both the SLA and DLP processes followed the ordering (from highest to lowest): sanding > soaking = rinsing = recovery = pouring. The maximum number concentration (see Table 3) for the SLA process was significantly higher for the sanding task compared with all other tasks ($p < 0.05$); for the DLP process, the order of tasks was (from highest to lowest): drying = printing > pouring ($p < 0.05$). Between printers, maximum particle concentrations for the DLP process were significantly higher compared with the SLA process for the drying and printing tasks, whereas maximum concentration for the SLA process was higher compared with the DLP process for the sanding task. For the SLA process, the GM particle size (1.34 μm) for the sanding task was significantly larger compared with all other tasks (range: 0.66 to 0.80 μm). For the DLP process, GM particle size for the sanding task (1.02 μm) was significantly larger compared with the drying (0.73 μm), pouring (0.70 μm), and printing (0.66 μm) tasks. Between processes, only the GM particle size differed for the sanding task, being significantly larger for the SLA process compared with the DLP process.

For maximum TVOC concentration (see Table 3), the SLA process yielded the following orderings (from highest to lowest): soaking = drying > curing > printing > pouring = recovery ($p < 0.05$) and rinsing = curing > printing > pouring = recovery ($p < 0.05$); or the DLP process, the ordering of maximum TVOC concentration among tasks (from highest to lowest) was soaking = drying > curing > pouring = printing = recovery = rinsing; $p < 0.05$. Between printers, maximum TVOC concentrations for the SLA process were significantly higher for the rinsing and printing tasks compared with the DLP process; $p < 0.05$.

For the DLP process, the concentration of 2-hydroxypropyl methacrylate was significantly higher for the recovery task compared with the curing task $p < 0.05$ (see Table 4). Among individual VOCs, only ethanol had sufficient data for statistical comparisons. For the SLA process, concentrations of ethanol were significantly higher for the recovery task compared with the printing task, and for the DLP process, concentrations were significantly higher for the recovery task compared with the curing task (see Table 4).

With the exception of Cr, Zn, and Zr the concentration of elements present in aerosols were generally not significantly different among tasks (see Supplemental Table S2). Concentrations of Cr were significantly higher for the pouring and recovery tasks compared with printing, curing, and sanding tasks. Concentrations of Zn were significantly higher for the pouring task compared with the printing task. For the DLP process, concentrations of Zr followed the ordering (from highest to lowest) pouring = recovery > sanding; $p < 0.05$.

## 4. Discussion

There is little understanding of task-based contributions to indoor air pollution and exposure potential from AM processes. In this study, two types of VP processes were evaluated on a task-by-task basis to improve knowledge of task durations and exposure potential. Previously, Runström Eden et al. collected work diaries and monitored exposures for material extrusion, material jetting, powder bed fusion, and VP processes and reported that the proportion of total exposure incurred during some tasks was not related to the percentage of time spent on these tasks during a workday [9]. Consistent with their observations, a comparison of Table 1 (task duration), Table 2 (real-time monitoring average concentrations), and Table 3 (real-time monitoring maximum concentrations) revealed that for these VP processes, task duration was not related to the magnitude of exposure potential. It was observed that the type of task influenced exposure potential. For example, during the SLA process, GM average and maximum TVOC concentrations exceeded 1000 and 9100 $\mu g/m^3$, respectively for the two-minute duration rinse task but were approximately 60 and 140 $\mu g/m^3$, respectively for the 145-min duration print task. Hence, the length of task duration was a poor surrogate for the magnitude of exposure potential, and caution is warranted to not discount the contribution of short-duration tasks to daily total exposure in exposure and risk assessments for VP processes. Additionally, each task in the VP process was performed differently, so to be most effective, controls might need to be tailored to each task.

### 4.1. Pouring

Average and maximum TVOC concentrations (PID data) did not exceed 5.5 $\mu g/m^3$ for this task. Hayes et al. monitored a pre-printing task of mixing resins outside of a VP printer and wiping excess resin away using rags wetted with IPA and acetone [24]. The authors stated that the "introduction" of resins (interpreted by the authors of the current study to mean placing a vat with mixed resin into a VP printer) yielded maximum TVOC concentrations (PID data) that ranged from approximately 3900 to 13,600 $\mu g/m^3$. It was unclear to what extent the use of IPA and acetone in their study influenced TVOC concentrations relative to off-gassing of the resins during mixing; however, the three to four orders of magnitude difference in TVOC concentrations reinforce the conclusion that task duration is a poor indicator of exposure potential and exemplifies how differences in the way that the same task is performed can yield dramatically different exposures.

### 4.2. Printing

Average particle concentrations (APS, CNC, and FMPS data) were below the background for both types of VP printers. This observation was consistent with Zhang et al. who reported that sub-micron size particle concentrations released during SLA printing were similar to the background [34]. In a series of studies, Väisänen et al. used a CNC to monitor emissions from several different VP machines and resins during printing [15,16]. In their studies, average particle concentrations during the operation of SLA and DLP

printers were approximately 1200 to 3600 #/cm$^3$. Differences in average concentrations between our and the Väisänen et al. studies is likely related to differences in printer design, print settings, resin properties (volatility, viscosity), and print surface area [35]. Hayes et al. used CNC and APS instruments to monitor emissions from an industrial-scale VP printer that used continuous liquid interface production (CLIP) technology; the average particle number concentration was indistinguishable from the background, though the CLIP machine had an internal filtration system [24]. Zisook et al. reported that for an industrial scale SLA printer operated in a room with 8.6 air exchanges per hour (ACH), CNC number concentrations were not distinguishable from the background [25].

Maximum particle number concentrations measured using a CNC were 125.0 and 270.5 #/cm$^3$ for the SLA and DLP printer, respectively; values were not significantly different. In a previous study, it was observed that, on average, DLP printers had significantly higher particle emission rates compared with SLA printers. Though particle emission rates differed on a group-level basis in that study, for the specific SLA and DLP printers used in the current study, particle number concentrations were not significantly different [23]. Väisänen et al. reported that maximum particle number concentrations measured using a CNC instrument while printing with SLA and DLP printers were approximately 4000 to 13,500 #/cm$^3$ [15,16]; however, VP particle concentrations tend to be lower than fused filament fabrication (FFF). As heat is used to melt the polymer, particle concentrations can reach a maximum of $2 \times 10^5$–$1 \times 10^6$ #/cm$^3$ [3]. Again, multiple factors such as differences in printer design, print settings, resin properties, and print surface area could explain the divergence in results among studies [35].

GM particle sizes during the print task (FMPS data) were 40.6 nm and 33.3 nm for the SLA and DLP printers, respectively. These values are in agreement with a previous study, in which GM particle sizes (FMPS data) were 41.3 nm and 28.8 nm for the same SLA and DLP printers, respectively [23]. Väisänen et al. used an FMPS to monitor particle size during the print task for several VP printers and reported GM sizes of 11 to 19 nm for an SLA printer and 19 to 45 nm for DLP printers [15], the latter of which is in general agreement with the current study.

The average TVOC concentration (PID data) for the SLA printer was 61.0 µg/m$^3$ whereas the concentration was below the background for the DLP printer. This concentration was much lower than the TVOC level (based on thermal desorption tube data) of approximately 5000 µg/m$^3$ previously reported for an SLA printer [34]. Zisook et al. reported that the average TVOC concentration (PID data) for an industrial scale SLA printer in a room with 8.6 ACH was not distinguishable from the background [25]. Other investigators have also reported low average TVOC concentrations during VP printing, though at levels that exceeded background. For example, Runström Eden et al. monitored VOC emissions from a desktop-scale SLA printer using thermal desorption tubes and reported that the TVOC concentration (sum of individual VOCs) was 140 µg/m$^3$, which is about a factor of two higher than in the current study [9]. Väisänen et al. also monitored VOC emissions from a desktop-scale SLA printer using thermal desorption tubes [15]. In a room with seven ACH, they reported average TVOC concentrations of 41 to 46 µg/m$^3$, which were similar to our results.

From the data presented in Table 3, the GM maximum TVOC concentration was 140.7 µg/m$^3$ for an SLA printer and 11.2 µg/m$^3$ for a DLP printer. In comparison, from their TD tube data, Väisänen et al. reported maximum TVOC concentrations of approximately 50 µg/m$^3$ for a desktop-scale SLA printer and 64 to 99 µg/m$^3$ for various DLP printers [15]. In another study, these authors reported maximum TVOC levels of 176 µg/m$^3$ and 427 µg/m$^3$ for SLA and DLP printers, respectively [16]. Vasilescu reported a similar maximum TVOC concentration of 360 µg/m$^3$ for a DLP printer [26]. While there is some difference in maximum TVOC concentrations amongst the current study data and existing literature, overall, results are within the same orders of magnitude despite differences in ventilation conditions of the study settings and printer-, feedstock-, and print-related factors.

Concentrations of acetone (evacuated canister data) during the printing task ranged from 2.7 to 10.1 $\mu g/m^3$ for the SLA printer and had a GM = 13.6 $\mu g/m^3$ for the DLP printer. These values were within the range previously reported for these printers (1.7 to 42.7 $\mu g/m^3$) that was measured using liquid impingers [23].

*4.3. Recovery*

Maximum particle number (CNC data) and TVOC (PID data) concentrations were slightly above the background during the resin recovery task. To our knowledge, there are no data on releases or exposures during the resin recovery task in the literature to which our results can be compared.

*4.4. Rinsing*

Based on the use of IPA, the potential for exposure to organic gases was the major concern for the rinsing task; for the SLA process (PID data), the GM average and maximum TVOC concentrations were 1042 $\mu g/m^3$ and 9113 $\mu g/m^3$, respectively. Yang and Li reported that for a combined ethanol cleaning/curing task, for objects made with an SLA printer, average and maximum TVOC concentrations (PID data) were 1775 $\mu g/m^3$ and 6180 $\mu g/m^3$, respectively [35]. Their results were on the same order of magnitude as measured in the current study, though it was unclear what proportion of their reported maximum TVOC concentration was attributed to ethanol cleaning (rinse) compared with the cure portion of this task. For the DLP process, the GM average and maximum TVOC concentrations for the rinse task were 0.4 $\mu g/m^3$ and 6.0 $\mu g/m^3$, respectively. Vasilescu reported that TVOC concentration reached approximately 10,000 $\mu g/m^3$ for a rinse task that involved objects made using a DLP printer [26]. Finally, Väisänen et al. reported an average TVOC concentration of approximately 11,000 $\mu g/m^3$ when VP-printed parts were rinsed with IPA over a sink [16]. It can be concluded that differences in the way that the same VP process task was performed dramatically influenced exposures.

The GM average (see Table 2) and maximum (see Table 3) TVOC measurements were significantly higher for objects made from an SLA printer compared with objects made from a DLP printer. Though task duration for the SLA printed parts was two minutes and for the DLP printed parts it was one minute, it does not fully explain the three orders of magnitude difference in concentrations. Given that the PID was calibrated before each task (see methods section), one possible explanation for the observed differences could be related to the time variation to place parts in the rinse containers. Even though the container was closed with a sealed lid before agitation, the time spent placing the part into IPA could explain the variability in the TVOC values that were observed from each printed part.

*4.5. Soaking*

Average particle concentrations during this task were similar to the background (Table 2), which is consistent with observations for an SLA process [34]. Based on PID measurements, GM average (approximately 8100 to 15,200 $\mu g/m^3$) and maximum (approximately 22,900 to 36,800 $\mu g/m^3$) TVOC concentrations were among the highest of all tasks, regardless of VP process. Though the TVOC metric is non-specific, the reported concentrations were ostensibly dominated by the use of IPA for this task [34]. Variability could likely reflect the manual nature of the process and it is likely levels will differ on how the operator performs the task. Hayes et al. reported that for a soak task, TVOC concentrations (PID data) were approximately 2750 to 10,100 $\mu g/m^3$ though that process used a proprietary fluorinated organic solvent [24]. Recently, Zhang et al. evaluated a soaking process for an SLA process that used IPA and reported that TVOC levels (thermal desorption tube data) were approximately 600 $\mu g/m^3$ [34]. This lower TVOC level could be related to task duration (10 min in the Zhang et al. study compared with 20 min in the current study), sampling methods (thermal desorption tubes compared with real-time PID), and/or configuration of the soaking tasks (e.g., air-tightness of containers). Again, task

duration can be a poor indicator of the magnitude of exposure potential, and differences in the way that the same VP process task is performed can yield dramatically different contributions to indoor air pollution and exposure potential.

*4.6. Drying*

Similar to the rinse and soak tasks, based on PID measurements, GM average (see Table 2) and maximum (see Table 3) TVOC concentrations were among the highest of all tasks, regardless of the VP process. To our knowledge, there are no data on exposure during the drying task in the literature to which we can compare our results; it is expected that the TVOC concentration corresponds to IPA concentration.

*4.7. Curing*

Average particle number concentrations (FMPS data) were below background during curing for both types of VP processes. Similarly, it was reported for an SLA curing task that average sub-micron particle concentrations were just 50 $\#/cm^3$ [34]. For both VP processes, maximum TVOC concentrations during the curing task were significantly higher than the pouring, printing, and recovery tasks and significantly lower compared with the soak and dry tasks (see Table 3). The pouring, printing, and recovery tasks involve resin, whereas the soaking and drying tasks involve IPA, but the curing task involves both resin and IPA. Zhang et al. measured TVOCs using thermal desorption tubes for a curing task and reported that levels were approximately 600 $\mu g/m^3$, which was a factor of three lower than in our study [34]. Several factors could explain this difference, including the type of resin, sampling approaches, and configuration of the cure station. In their study, IPA emissions were highest during soaking, lower during curing, and not detected during printing [34]. As mentioned previously, Yang and Li monitored a combined ethanol cleaning/curing task and for objects made with an SLA printer, average and maximum TVOC concentrations (PID data) were 1775 $\mu g/m^3$ and 6180 $\mu g/m^3$, respectively [35]. In the current study, for the SLA process, GM average TVOC concentration (1867.2 $\mu g/m^3$) was similar, though GM maximum TVOC concentration (3083.4 $\mu g/m^3$) was a factor of two lower.

*4.8. Sanding*

The final task in the VP process was sanding printed objects to remove surface defects where the build supports were cut off. Given the low-energy input, but abrasive nature of sanding, it was expected that particles released during this task would be micronscale. In a study by Bressot et al., some particles collected on a transmission electron microscopy grid following sanding of objects made using an SLA printer had quasi-spherical shapes and sizes of 1 µm or larger [36]. Consistent with this expectation, many particles that were observed in our SEM images had similar micronscale characteristics (see Supplemental Figure S8). Bressot et al. also described quasi-spherical particles with sizes of 120 nm diameter. Though not shown in the Supplemental File Figure S8, we also observed particles with similar size and morphology during SEM analyses. Consistent with the microscale nature of aerosol particles observed in SEM images, particle size (APS data) during the sanding task were micronscale particles. The average GM size for particles released during sanding of objects made using an SLA printer was significantly larger than objects made using a DLP printer (see Table 2). This observation was somewhat surprising given that both VP processes used the same resin and the same sandpaper grits. This difference could be due to how the printer manufacturer's software generates their support on the part to be printed. The supports were generated based on the recommended settings for each printer, so the part of the support that secures the initial layer of the actual part will vary in size and shape, as well as the number of supports used for each build. When using the flush cutters to remove the supports from the part, there were instances that the connecting part of the support could not get cut as close to the initial layer of the object, which meant different height protrusions were sanded, and in turn, could influence particle size. It is important to note the rest of the printed object would not require as much

sanding as a FFF part. FFF printed parts are sanded to help improve surface quality and properties [37].

### 4.9. Relationships among Tasks

Maximum TVOC concentrations were significantly higher for the soaking, drying, and curing tasks compared with the printing task. Other studies have also documented that specific tasks in VP processes other than printing can confer higher exposure to organic gases [9,16,24,26,35]. Collectively, our results and existing literature support the conclusion that for VP processes the printing task does not necessarily contribute to the highest level of indoor air pollution or infer the highest exposure potential as the operator does typically leave the area during this task. As such, consideration for contaminant releases and exposure potential throughout the entire VP process is essential as part of any risk mitigation strategy.

In the current study, gas-phase releases presented an exposure potential of concern, which is consistent with previous publications [9,15,16,24,26,35]. For both types of VP processes, TVOC concentrations for the soaking and drying tasks (and maximum TVOC for the SLA process rinsing task) were significantly higher compared with the pouring, printing, and recovery tasks. Though the magnitude of TVOC concentrations differed, consideration must also be given to the composition of gas-phase contaminants released into indoor air during these tasks. The rinsing, soaking, and drying tasks all involved use of IPA, which is an eye, nose, and throat irritant with a boiling point of 82.7 °C [38]. Given its relatively low boiling point, IPA has high volatility and will readily evaporate into the gas phase. In contrast, the pouring, printing, and recovery tasks all involved a resin, which has a boiling point >100 °C [39]. Given this relatively higher boiling point compared with IPA, it is expected that the resin will have lower volatility, and therefore, release VOCs at lower concentrations. TVOC concentrations during the curing task were lower than the rinsing, soaking, and drying tasks but greater than the pouring, printing, and recovery tasks, which likely reflects that this task involved heating printed objects with residual IPA and under- or uncured resin. Analysis of charcoal tube samples (see Table 4) revealed that 2-hydroxypropyl methacrylate, a known sensitizer [14], was released at varying concentrations during the pouring, printing, recovery, and curing tasks. The highest concentrations of this sensitizer were measured during the resin recovery task. Pouring of the resin required little time as it involved adding liquid to an open-top vat immediately followed by closing the printer cover. In contrast, resin recovery required more time as it involved pouring resin from the vat into a narrow-mouthed bottle. Additionally, during recovery, the resin was agitated using a putty knife to remove residual resin from the vat, which could contribute to the volatilization of this substance. Both the printers and the curing station machines have built-in enclosures, which could have helped to reduce airborne concentrations of 2-hydroxypropyl methacrylate. Zhang et al. reported that 2-hydroxypropyl methacrylate was among the highest emitting chemicals during SLA tasks; emission rates decreased as the process went from printing to soaking to curing [34]. Though we did not monitor for this compound during the soaking task, our results for the SLA process are consistent in that concentrations decreased from the printing to the curing tasks (Table 4). Canister samples quantified acetaldehyde, designated by NIOSH as a potential occupational carcinogen [38], and acetone, a central nervous system depressant [38], in the air during the pouring, printing, recovery, and curing tasks. According to the International Agency for Research on Cancer, acetaldehyde is classified as a group 2B carcinogen [40]. Methylene chloride, designated by NIOSH as a potential occupational carcinogen [38], was quantified in the air during the pouring, printing, and recovery tasks. The presence of methylene chloride was observed in all samples during the pour, print, and recovery task using the DLP printer; however, none of these concentrations were significantly different from one another. Previously, Väisänen et al. monitored specific VOC emissions during the print task for SLA and DLP printers using various resins [15,16]. For the SLA print task, they reported that major VOC emissions included acrylates such

as 2-hydroxypropyl methacrylate (as well as methyl methacrylate and ethyl methacrylate, both of which were below analytical LODs in the current study) and carbonyls such as acetaldehyde and acetone. For the DLP print task, they quantified acrylates such as methyl methacrylate as well as acetaldehyde and acetone. Zhang et al. reported that 88% of the sum of VOCs measured on thermal desorption tubes during SLA printing were attributed to 2-hydroxypropyl methacrylate, propylene glycol, 2-hydroxyethyl methacrylate, acetone, and crotonic anhydride [34].

Seventeen elements were quantified on one or more filter samples during the task simulations (see Supplemental File). Among these 17 elements, five (As, Sn, Ti, Zn, and Zr) were quantified in both the bulk resin and air samples and four (Ba, Cd, Fe, and Mo) were below analytical LODs for the bulk resin analyses but quantified on air samples. Many photoinitiators are metal-based molecules. Several elements used as photoinitiators that were also quantified in bulk resin and/or air samples included Zn, Fe, Ti, and Ba compounds [11,17–20]. Concentrations of Cr were significantly greater for the pouring and recovery tasks compared with the printing and curing tasks and concentrations of Zn were significantly higher for the pouring task compared with the printing task. Given that the same resin was used for both VP processes, these observations support the premise that both process technology (SLA or DLP) and task influence the release of metals. For example, initially, we thought it could be that the physical movement of the bulk liquid resin during pouring generated more aerosol than during printing; however, particle concentration on a number-basis was significantly higher during the printing task (see Table 3). An alternate explanation could be that particle size, and therefore mass, was larger for particles released during pouring (FMPS data for DLP process) compared with the printing, recovery, rinsing, soaking, and drying tasks (see Supplemental Table S2), which could have aerosolized more Cr and Zn.

*4.10. Limitations*

Due to the ever growing 3-D printing market and different ways to perform certain tasks, there are potential limitations of this study worth noting. Only one resin was evaluated for these processes. There are numerous VP resins currently available on the commercial market, because of this, it is anticipated that resin properties (e.g., viscosity and boiling temperature of ingredients) will differ among manufacturers, which in turn will influence exposure potential. Additionally, this resin was discontinued by the manufacturer during our study, so we had a limited supply. As a result, the resin that was recovered during one run was then used at the start of the next run, along with any additional "fresh" resin that remained. Though this practice of combining recovered resin and fresh resin is common industrial practice, it is not standardized and could have introduced variability to exposure potential among tasks. The resin pouring, recovery, and sanding tasks were performed manually for each run. This could introduce variability in results. For example, how quickly resin was poured into vats or recovered into containers or the amount of force during sanding will likely vary with the person performing the task.

## 5. Conclusions

A task-by-task characterization of two types of VP processes, SLA and DLP, demonstrated that particles and gases were released during all tasks. Major themes that emerged from this study were: (1) time spent on a task did not relate to the magnitude of potential exposure; (2) the specific method used to perform a task greatly impacted potential exposure; (3) the print task was not always the highest exposure task; and (4) the composition of organic gases, and potential hazard from exposure, varied among tasks. Careful consideration of all tasks is important for the development of exposure mitigation strategies and control technologies (e.g., ventilation, automation of tasks) to minimize indoor air pollution and exposure potential from VP processes.

**Supplementary Materials:** The following supporting information can be downloaded at: https://www.mdpi.com/article/10.3390/buildings12081222/s1, Table S1: Elemental content of bulk feedstock grey resin; Table S2: Background-corrected concentrations of aerosol elemental constituents for select tasks; Figure S1: Untransformed average contaminant concentrations by process and task (real-time data); Figure S2: Untransformed maximum contaminant concentrations by process and task (real-time data); Figure S3: Untransformed average contaminant concentrations by process and task (specific VOCs and elements); Figure S4: Electron microscopy images and energy dispersive X-ray spectra of particles from pouring; Figure S5: Electron microscopy images and energy dispersive X-ray spectra of particles from printing; Figure S6: Electron microscopy images and energy dispersive X-ray spectra of particles from recovery; Figure S7: Electron microscopy images and energy dispersive X-ray spectra of particles from curing; Figure S8: Electron microscopy images and energy dispersive X-ray spectra of particles from sanding.

**Author Contributions:** Conceptualization, L.N.B., A.B.S. and A.K.K.; Methodology, L.N.B., A.B.S., S.B.M.J. and A.K.K.; Formal Analysis, L.N.B., A.B.S. and M.A.V.; Investigation, L.N.B., A.B.S. and A.K.K.; Data Curation, L.N.B., A.B.S., R.F.L., A.C.R., D.A.B., M.A.V.; Writing—Original Draft Preparation, L.N.B. and A.B.S.; Writing—Review & Editing, L.N.B., A.B.S., A.K.K., R.F.L., S.B.M.J., A.C.R., D.A.B. and M.A.V.; Visualization, L.N.B. and A.B.S.; Supervision, A.B.S., R.F.L. and M.A.V.; Project Administration, A.B.S.; Funding Acquisition, A.B.S., R.F.L. and M.A.V. All authors have read and agreed to the published version of the manuscript.

**Funding:** This research received no external funding.

**Institutional Review Board Statement:** Not applicable.

**Informed Consent Statement:** Not applicable.

**Data Availability Statement:** The data presented in this study are available on request from the corresponding author.

**Acknowledgments:** The authors thank E. Glassford and G. Roth at NIOSH for critical review of this manuscript prior to submission to the journal. We also thank J. Roberts (NIOSH) for helpful guidance with the air sampling method selection. The findings and conclusions in this report are those of the authors and do not necessarily represent the official position of the National Institute for Occupational Safety and Health, Centers for Disease Control and Prevention.

**Conflicts of Interest:** The authors declare no conflict of interest.

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
