# Peer review of "Potential for Exposure to Particles and Gases throughout Vat Photopolymerization Additive Manufacturing Processes"

_buildings, doi:10.3390/buildings12081222_

Round 1
Reviewer 1 Report
The manuscript describes determination of particles and VOCs in vat photopolymerization (VP) process. The potential exposure for these compounds were well evaluated for each task. The manuscript is well written and includes sufficient novel information.
On the scientific paper, any results in the manuscript must be replicable based on the method section by another researchers. However, in this paper, description about measuring method is insufficient. Especially, information about GC-MS analysis (type of instrument, column, separation condition, sample preparation conditions, used m/z, etc.
In my opinion, unit should be described in the table (not table title). In Table 1, "Duration time (min) should be added in row 1. Similarly, in Table 4, "Compound (μg/m3)" should be added in the table.
Error bars demonstrated in the supplemental figures for VOC concentration are very big. Why the repeatability for VOC measurement is so insufficient?
An abbreviation that already defined should be used without definition. In Table 3, the definition for CNC, FMPS... would not needed. They were defined in Table 2.
Author Response
Thank you for taking the time to review our manuscript. I included the responses to your comments in the attached word document.

Reviewer 2 Report
First of all, I would like to congratulate the author for their work that I've read with great interest. I however have few comments:
- Line 376: usually SLA and in particular DLP customer level printers should not be used without the huge protective lid open. Maybe I missed this detail in the document, but how did you used the printers, with or without lid? because the printing process is creating heat nearby the screen and the resin tank which I think increase the levels of pollutant. I guess the results are thus different for the measurement with and without the lid on.
- Line 443: why are you including the sanding process as you've chosen the printing method that is most used because the final object do not need sanding but maybe only painting?
- Line 694-697: I think that a few sentences on FDM 3D printer are missing in the context or at the end of the discussion. In fact, FDM have more impact on indoor air during the printing process than the pre and post printing steps as the "plastic" is heated and extruded at high temperature. Sanding is maybe also more important for FDM than for VP processes.
Author Response

(The authors gave the same response as above.)
